# Could We Predict the Response of Immune Checkpoint Inhibitor Treatment in Hepatocellular Carcinoma?

**DOI:** 10.3390/cancers14133213

**Published:** 2022-06-30

**Authors:** Choong-kun Lee, Stephen L. Chan, Hong Jae Chon

**Affiliations:** 1Division of Medical Oncology, Department of Internal Medicine, Yonsei University College of Medicine, Seoul 03722, Korea; cklee512@yuhs.ac; 2State Key Laboratory of Translational Oncology, Department of Clinical Oncology, Sir YK Pao Centre for Cancer, Prince of Wales Hospital, The Chinese University of Hong Kong, Hong Kong, China; 3Medical Oncology, Department of Internal Medicine, CHA Bundang Medical Center, CHA University, Seongnam 13496, Korea

**Keywords:** hepatocellular carcinoma, anti-programmed cell-death protein (ligand)-1, immune checkpoint inhibitor, predictive biomarker, clinical biomarker, translational biomarker

## Abstract

**Simple Summary:**

The use of anti-programmed cell-death protein (ligand)-1 (PD-[L]1) is now a standard of care for treating hepatocellular carcinoma (HCC). However, the treatment only benefits 10–20% of patients when used as a monotherapy. The unique environments of hepatitis and/or cirrhosis, which continuously interact with the hosts’ immune systems, make it difficult to find appropriate biomarkers to predict the response or lack of response of anti-PD-1/PD-L1 treatment in HCC. The current review aimed to present both clinical and translational biomarkers for anti-PD-1/PD-L1 treatment in HCC.

**Abstract:**

The use of anti-programmed cell-death protein (ligand)-1 (PD-[L]1) is an important strategy for treating hepatocellular carcinoma (HCC). However, the treatment only benefits 10–20% of patients when used as a monotherapy. Therefore, the selection of patients for anti-PD-1/PD-L1 treatment is crucial for both patients and clinicians. This review aimed to explore the existing literature on tissue or circulating markers for the identification of responders or non-responders to anti-PD-1/PD-L1 in HCC. For the clinically available markers, both etiological factors (viral versus non-viral) and disease extent (intra-hepatic vs. extrahepatic) impact the responses to anti-PD-1/PD-L1, warranting further studies. Preliminary data suggested that inflammatory indices (e.g., neutrophil-lymphocyte ratio) may be associated with clinical outcomes of HCC during the anti-PD-1/PD-L1 treatment. Finally, although PD-L1 expression in tumor tissues is a predictive marker for multiple cancer types, its clinical application is less clear in HCC due to the lack of a clear-cut association with responders to anti-PD-1/PD-L1 treatment. Although all translational markers are not routinely measured in HCC, recent data suggest their potential roles in selecting patients for anti-PD-1/PD-L1 treatment. Such markers, including the immune classification of HCC, selected signaling pathways, tumor-infiltrating lymphocytes, and auto-antibodies, were discussed in this review.

## 1. Introduction

Hepatocellular carcinoma (HCC) accounts for over 80% of primary liver cancer [1,2]. Typically, more than 80% of HCCs occur in the background of cirrhotic liver, which is characterized by long-standing inflammation due to viral hepatitis or metabolic or chemical injury [3,4]. HCC is highly lethal due to its delayed presentation, resistance to drug treatment, and underlying hepatic decompensation [5,6]. The mainstay of systemic therapy for HCC has been muti-targeted tyrosine kinase inhibitors (TKIs), such as sorafenib, lenvatinib, regorafenib, and cabozantinib [7,8,9,10]. TKIs typically lead to disease control or result in modest response for a period of time; however, resistance to TKIs is inevitable in most patients after a few months of treatment.

Recent advances in immune checkpoint inhibitors (ICIs) have, however, changed the above scenario. ICIs, particularly the anti-programmed cell death protein (PD)-1/ligand (PD-L1) antibodies, can potentially reverse the immune-exhausted microenvironment of HCC and induce cytotoxic T cell-mediated destruction of HCC [11]. In clinical trials, monotherapy using anti-PD1 was associated with a radiological response rate of 10 to 20% in patients with HCC, including both complete and partial responses [12,13]. The responses can potentially be durable in some patients, thus explaining the observation of a plateau, frequently known as the tail, in the Kaplan–Meier survival curves of clinical trials. However, the initial high expectations from ICIs were disappointed by the failures of phase III clinical trials on anti-PD-1 monotherapy to reach the primary objectives of improved overall survival (OS) compared to sorafenib [14,15]. The negative results of clinical trials on monotherapy using anti-PD-1 could be explained in multiple ways, including the use of subsequent therapy, heterogeneity of patients with HCC, the lack of useful predictive biomarkers for patient selection, and accelerated progression and neutralizing auto-antibodies [16,17,18].

Strategically, there are two approaches to improve the outcome of anti-PD1/PDL-1 treatment in HCC. First, combinational treatment of anti-PD-1/PD-L1 with other ICIs or targeted agents could be synergistic, thereby significantly enhancing the treatment outcomes in patients. The approval of the atezolizumab–bevacizumab combination as the first-line treatment in HCC is the first notable example [19]. Recently, another combinational regimen of tremelimumab–durvalumab was shown to improve the median OS over that of sorafenib treatment in a phase III clinical trial [20]. The second approach is to develop methodologies to select patients who are more likely to derive benefits from the anti-PD-1/PDL-1 treatment. Experiences with other cancer types suggested that patients could be enriched by clinical biomarkers to improve the outcomes of anti-PD-1/PD-L1 treatment. For example, the phenotype of deficit mismatch repair (d-MMR) is a tumor-agnostic marker that predicts high responses and prolonged survival in response to ICI treatment in different cancers [21]. In lung cancer, the high immunohistochemical staining of PD-L1 in tumor tissues is known to be associated with clinical benefits in anti-PD-1 treatment [22]. For HCC, robust studies were conducted by different groups to identify markers predictive of benefits or resistance to anti-PD-1/anti-PD-L1 treatment. However, the overall picture is more complex in HCC than in other solid tumors due to the unique environment of hepatitis and/or cirrhosis, which continuously interacts with the hosts’ immune systems. The current review aimed to present both clinical (Table 1) and translational (Table 2) biomarkers for anti-PD-1/PD-L1 treatment in HCC.

## 2. Clinical Biomarkers

### 2.1. Etiology

Chronic infection with hepatitis B virus (HBV) and hepatitis C virus (HCV) is a traditional major risk factor associated with HCC [23]. The virus-associated mechanisms that cause liver cancer are complex, and HCC develops mostly in cirrhotic liver (about in 90% of the cases), whereas HCC development in the normal liver is a rare event (less than 10% of the cases) [24]. HBV infections account for 75–80% of virus-associated HCCs, and the integration of genetic material of HBV into the human genome leads to p53 inactivation, inflammation, or activation of various oncogenic pathways, including PI3K/Akt/STAT3 pathway and Wnt/b-catenin (induction of oxidative stress), which induce hepatocarcinogenesis [25,26,27,28]. Unlike that in HBV infection, the genetic material of HCV is not integrated into the host’s genome; rather, the HCV proteins induce chronic inflammation, which leads to the development of HCC [28,29]. In addition to viral causes, fatty liver disease, especially non-alcoholic fatty liver disease (NAFLD) [30], which includes non-alcoholic steatohepatitis (NASH), is the fastest-growing etiology due to lifestyle changes in the western dietary pattern, increased obesity, and improved antiviral therapy [31]. Multiple mechanisms, including steatosis-induced necroinflammation, the release of inflammatory cytokines, and immune microenvironment alterations, are the key driving forces in NAFLD-associated HCC [32,33].

A recent meta-analysis [16] evaluated the effect of etiology in terms of efficacy across three large randomized controlled phase III trials of immunotherapies for HCC, namely anti-PD-L1 in combination with anti-vascular endothelial growth factor (VEGF) (IMbrave150 [19]), or anti-PD-1 monotherapy (CheckMate 459 [14]) compared to sorafenib, or second-line anti-PD-1 monotherapy compared to placebo (Keynote-240 [15])-treated patients. In this large meta-analysis (total *n* = 1656), patients with HBV-related HCC and HCV-related HCC showed superior survival benefits from immunotherapy than the control, although it was not so in patients with non-viral HCC. Among the additional validation cohort with HCC patients treated with anti-PD/PD-L1, NAFLD was independently associated with shortened survival of patients with HCC after anti-PD-1/PD-L1 treatment. Preclinical evidence showed that NASH progression is associated with increased activated CD8^+^PD1^+^T cells; anti-PD-1 treatment did not lead to tumor regression, indicating that tumor immune surveillance was impaired. A recent preclinical study suggested that an anti-PD1 and CXCR2 inhibitor combination selectively reprograms tumor-associated neutrophils from a pro-tumor to an anti-tumor phenotype that can overcome the resistance of NASH-HCC to anti-PD1 therapy [34]. In the recent HIMALAYA [35] phase III trial, testing the combination of anti-CTLA4 and anti-PD-L1 inhibitors for first-line treatment of advanced HCC, patients with HBV-related or non-viral-etiology HCC were benefitted in terms of OS, compared to those receiving sorafenib, although it was not so in cases of HCV-related HCC. Further investigation would be required for such contradictory results.

### 2.2. Disease Extent

Treatment options for HCC are dependent on the stage (Barcelona clinic liver cancer, BCLC, staging system [36]) of the disease. Patients in the intermediate stage (BCLC-B) and advanced stage (BCLC-C) are candidates for systemic treatment. In recent first-line phase III trials for advanced HCC (IMbrave150, HIMALAYA), atezolizumab and bevacizumab or tremelimumab and durvalumab were reported to be superior to sorafenib in terms of OS of patients with BCLC-C, although not for those with BCLC-B [19,35]. However, in a Chinese phase III trial conducted mostly for patients with B-viral HCC, those with BCLC-B or BCLC-C also benefitted from anti-PD-1 and anti-VEGF treatments relative to that from sorafenib treatment [37]. Patients in the BCLC-C stage presented with vascular invasion or extrahepatic spread. In IMbrave150 and HIMALAYA trials, patients with extrahepatic spread achieved OS benefit from the first-line atezolizumab and bevacizumab or tremelimumab and durvalumab treatment than from sorafenib treatment. Anti-tumor immune response to ICIs differs in an organ-specific manner [38], and liver metastasis is associated with poor response to immunotherapy monotherapy. Accordingly, intra-hepatic tumors of HCC were reported to possibly be less responsive to immunotherapy monotherapy than extrahepatic lesions [39,40]. Preclinical evidence also supported that liver tumors show reduced peripheral T cell numbers and diminished tumoral T cell diversity and function, creating an immune desert. Yu et al. showed that in mouse models, liver-directed radiotherapy could eliminate immunosuppressive hepatic macrophages, enhancing the anti-tumor effect of immunotherapy [41]. Further strategies would be required to enhance the anti-tumor effect of ICIs in intrahepatic lesions of patients with advanced HCC, along with the combination of local control.

Tumoral macrovascular invasion (MVI) of hepatic and/or portal vein branches is a common phenomenon in advanced HCC and is usually associated with a poorer prognosis than HCC without MVI. Patients with HCC and MVI, including Vp4 (presence of a tumor thrombus in the main trunk and/or contralateral portal vein), show superior survival when treated with atezolizumab and bevacizumab (anti-VEGF) than with sorafenib [19,42]; however, there was no additional survival benefit in durvalumab and tremelimumab treatment compared to that in sorafenib treatment in a subgroup analysis of the HIMALAYA trial [35]. Further studies (translational and clinical) would be required to investigate whether anti-VEGF treatment in combination with immunotherapy has any additional benefit in HCC with MVI.

### 2.3. Laboratory Tests

In daily practice, we performed laboratory tests to examine patient status; some features from laboratory (blood) tests can be used as biomarkers to predict immunotherapeutic efficacy in patients with advanced HCC and low invasiveness.

Elevated tumor markers, especially α-fetoprotein (AFP), are considered prognostic markers for poor clinical outcomes among patients with HCC. Recent randomized phase III studies showed contradictory results in terms of the benefit of immunotherapy compared to that in the control. In the first-line phase III CheckMate 459 [14] and HIMALAYA [35] trials, patients with high baseline AFP levels (≥400 ng/mL) achieved longer OS when treated with immunotherapy rather than sorafenib. However, results of the IMbrave150 study showed that patients with low baseline AFP levels (AFP < 400 ng/mL) were associated with longer OS and PFS when treated with immunotherapy rather than sorafenib [19]. Since AFP level is related to the tumor or patient characteristics, interpretation should be performed with caution. As generally seen in other treatments, a decline in post-treatment tumor marker level is associated with better efficacy of immunotherapy in advanced HCC; the AFP response at 6 weeks after atezolizumab plus bevacizumab initiation especially seemed to be a potential surrogate biomarker for prognosis [43,44,45].

The usage of circulating immune cells as predictive biomarkers for immunotherapy was extensively investigated. Contrary to specific immune cells that require additional experiments to obtain, we can inexpensively and reproducibly obtain information about complete blood cell differential counts from the patients in daily laboratory tests. A neutrophil-to-lymphocyte ratio (NLR), defined by the ratio of an absolute number of neutrophils to that of lymphocytes, is an especially well-known marker for selecting patients that are benefitted from immunotherapy in various tumor types [46]. A correlation was reported between circulating neutrophils and neutrophils in the tumor microenvironment, and low circulating lymphocyte levels were associated with low levels of tumor-infiltrating lymphocytes (TILs), thereby resulting in reduced anti-tumor T-cell responses [47,48]. In addition to NLR, platelet-to-lymphocyte ratio (PLR) is regarded as a biomarker of immunotherapy response since platelets are also part of an inflammatory process [49]. In the CheckMate 040 study, OS benefit was observed in patients with low NLR or PLR tertile due to nivolumab treatment than in others. Other studies also reported the predictive role of NLR or PLR in immunotherapy of advanced HCC [44,50]. Kim et al. reported that elevated NLR could predict the occurrence of hyper-progressive disease and inferior survival rate after anti-PD-1 blockade [51]. However, since NLR is also an independent prognostic factor for patients with HCC treated with sorafenib [52,53], further studies would be required to confirm the role of NLR or PLR in patients with HCC, to clarify whether it is a prognostic biomarker for the general population or whether there is any specific role by which it can identify patients with maximum possible benefit from immunotherapy than from tyrosine kinase inhibitor therapy.

### 2.4. PD-L1 Expression

PD-L1 is widely expressed on the surface of tumor cells, and its high expression in the tumor microenvironment is generally regarded as a biomarker for anti-PD-1/PD-L1 immunotherapy in various tumors, especially in NSCLC [22,54,55]. In HCC, PD-L1 expression was reported to be approximately 10 to 20% in tumor cells [14,56], and PD-L1 expression in HCC tumor cells is considered to be associated with tumor aggressiveness and poor survival [57]. Several clinical trials evaluated whether PD-L1 expression has predictive value as a biomarker for immune checkpoint inhibitor efficacy in patients with HCC. However, the types of PD-L1 antibodies for immunohistochemistry (28-8, 22C3, SP142, SP263) and the way of interpretation vary across trials, and the determination of their roles has been challenging. Among the patients treated with anti-PD-1 monotherapy, PD-L1 positive HCC, whether in tumor cells or tumor and immune cells combined, seem to respond better than those with negative PD-L1 expression [12,13,58]. Genomic analyses for the phase I trial of atezolizumab and bevacizumab in patients with HCC reported that high expression of PD-L1, as per RNA-seq, is related to better response and longer PFS [59]. In recent phase III trials, atezolizumab plus bevacizumab showed benefit over sorafenib in terms of PFS for a tumor or immune cells in PD-L1-positive patients [60]. However, since the recent phase III HIMALAYA trial showed the benefit of doublet immunotherapy over that of sorafenib, regardless of PD-L1 expression [35], further investigation is warranted in this regard.

**Table 1 cancers-14-03213-t001:** Clinical biomarkers.

Factor	Detail	Outcome	Regimen	Line of Treatment	Trial (Phase)	Ref.
Etiology	Hepatitis B	OS (HR 0.51) and PFS (HR 0.47) benefit	Atezolizumab + Bevacizumab vs. Sorafenib	1st	IMbrae150 (III)	Finn et al. NEJM 2020 [19]
	Hepatitis C	OS (HR 0.43) benefit	Atezolizumab + Bevacizumab vs. Sorafenib	1st	IMbrae150 (III)	Finn et al. NEJM 2020 [19]
	Hepatitis B	OS (HR 0.64) benefit	Durvalumab + Tremelimumab vs. Sorafenib	1st	Himalaya (III)	Abou-alfa et al. NEJM Evidence 2022 [20]
	Non-viral	OS (HR 0.74) benefit	Durvalumab + Tremelimumab vs. Sorafenib	1st	Himalaya (III)	Abou-alfa et al. NEJM Evidence 2022 [20]
	HBV	OS (HR 0.57) benefit	Pembrolizumab vs. Placebo	2nd	KEYNOTE-240 (III)	Finn et al. JCO 2019 [15]
	HBV	OS benefit	Nivolumab, Atezolizumab + Bevacizumab, Pembrolizumab (Meta-analysis)	1st–2nd	CheckMate-459, IMbrave150 and KEYNOTE-240 (III)	Pfister et al. Nature 2021 [16]
	HCV	OS benefit	Nivolumab, Atezolizumab + Bevacizumab, Pembrolizumab (Meta-analysis)	1st–2nd	CheckMate-459, IMbrave150 and KEYNOTE-240 (III)	Pfister et al. Nature 2021 [16]
	NAFLD	Worst survival	Nivolumab, Atezolizumab + Bevacizumab, Pembrolizumab (Meta-analysis)	1st–2nd	CheckMate-459, IMbrave150 and KEYNOTE-240 (III)	Pfister et al. Nature 2021 [16]
BCLC stage	BCLC C (no benefit for BCLC B)	OS (HR 0.58) and PFS (HR 0.58) benefit	Atezolizumab + Bevacizumab vs. Sorafenib	1st	IMbrae150 (III)	Finn et al. NEJM 2020 [19]
	BCLC C (no benefit for BCLC B)	OS (HR 0.76) benefit	Durvalumab + Tremelimumab vs. Sorafenib	1st	Himalaya (III)	Abou-alfa et al. NEJM Evidence 2022 [20]
	BCLC B and C	OS and PFS benefit	Sintilimab + bevacizumab biosimilar vs. Sorafenib	1st	ORIENT-32 (III)	Ren et al. Lancet Oncol. 2021 [35]
Extrahepatic Spread	Extrahepatic spread	OS (HR 0.5) benefit	Atezolizumab + Bevacizumab vs. Sorafenib	1st	IMbrae150 (III)	Finn et al. NEJM 2020 [19]
	Extrahepatic spread	OS (HR 0.67) benefit	Durvalumab + Tremelimumab vs. Sorafenib	1st	Himalaya (III)	Abou-alfa et al. NEJM Evidence 2022 [20]
Macrovascular invasion	Macrovascular invasion	OS (HR 0.58) benefit	Atezolizumab + Bevacizumab vs. Sorafenib	1st	IMbrae150 (III)	Finn et al. NEJM 2020 [19]
	No macrovascular invasion	OS (HR 0.77) benefit	Durvalumab + Tremelimumab vs. Sorafenib	1st	Himalaya (III)	Abou-alfa et al. NEJM Evidence 2022 [20]
Tumor Marker	AFP < 400ng/mL	OS (HR 0.52) and PFS (0.49) benefit	Atezolizumab + Bevacizumab vs. Sorafenib	1st	IMbrae150 (III)	Finn et al. NEJM 2020 [19]
	AFP ≥ 400ng/mL	OS (HR 0.64) benefit	Durvalumab + Tremelimumab vs. Sorafenib	1st	Himalaya (III)	Abou-alfa et al. NEJM Evidence 2022 [20]
	AFP ≥ 400ng/mL	OS benefit	Nivolumab vs. Sorafenib	1st	CheckMate-459 (III)	Yau et al. Lancet Oncol. 2022 [14]
	AFP < 400ng/mL	OS benefit	Nivolumab	1st–2nd	CheckMate-040 (I/II)	Sangro et al. J. Hep. 2020 [61]
	AFP < 200ng/mL	OS (HR 0.68) and PFS (HR 0.64) benefit	Pembrolizumab vs. Placebo	2nd	KEYNOTE-240 (III)	Finn et al. JCO 2019 [15]
Other laboratory tests	Neutrophil-to-lymphocyte ratio	OS benefit for pts with low tertile	Nivolumab	1st–2nd	CheckMate-040 (I/II)	Sangro et al. J. Hep. 2020 [61]
	Platelet-to-lymphocyte ratio	OS benefit for pts with low tertile	Nivolumab	1st–2nd	CheckMate-040 (I/II)	Sangro et al. J. Hep. 2020 [61]
PD-L1 IHC	PD-L1 TC (28-8) ≥ 1%	No significant benefit	Nivolumab vs. Sorafenib	1st	CheckMate-459 (III)	Yau et al. Lancet Oncol. 2022 [14]
	PD-L1 TC (28-8) ≥ 1%	ORR (28% vs. 16%) and OS (28.1 vs. 16.6 months, *p* = 0.032) benefit	Nivolumab	1st–2nd	CheckMate-040 (I/II)	El-Khoueiry et al. Lancet 2017 [12]
	PD-L1 CPS (22C3) ≥ 1%	ORR (32% vs. 20%, *p* = 0.021) benefit	Pembrolizumab	2nd	KEYNOTE-224 (II)	Zhu et al. Lancet Oncol. 2018 [13]
	PD-L1 TPS (SP142) ≥ 1%	ORR 36% vs. 11%	Camrelizumab	2nd	NCT02989922 (II)	Qin et al. Lancet Oncol. 2020 [14]
	PD-L1 TC or IC (SP263) ≥ 1%	PFS (OR 2.69) benefit	Atezolizumab + Bevacizumab vs. Sorafenib	1st	IMbrae150 (III)	Cheng et al. J. Hepatol. 2022 [58]

## 3. Translational Biomarkers

### 3.1. Immune-Specific Class of HCC

Daniela et al., previously characterized patients with high immune infiltration and molecular features resembling melanoma who responded to ICIs, as the immune class of HCC (approximately 25% of patients) [62]. Recently, Carla et al. further dichotomized the immunogenomic classification of HCC into inflamed and non-inflamed tumors [63]. However, their analyses were not based on NGS data of advanced patients who had received ICI treatment but rather on the results of pathology and immunohistochemical analyses to evaluate the correlation between expression patterns and the presence of both immune cell infiltrates and immune regulatory molecules. Therefore, the predictive capacity of such classification would need further investigation in patients receiving immunotherapy.

### 3.2. Tumor-Infiltrating Lymphocytes (TILs) and T-Cell Inflamed Gene Expression Profiles (GEP)

Tumor-infiltrating lymphocyte density and phenotypes are good predictive indicators of better responses to immunotherapy [61,64,65,66]. In the exploratory analysis of the CheckMate 040 trial [67], improved OS of patients with HCC who were being treated with nivolumab correlated with higher densities of CD3^+^ or CD8^+^ TILs. Gene expression, known to be related to immune cytolytic activity, was also demonstrated to be associated with the clinical outcome of certain tumors after checkpoint blockade treatment [68,69]. Recently, a T cell-inflamed gene expression profile (GEP) was presented as a predictive indicator of response to anti-PD-1-based therapy [70]. In the CheckMate 040 trial [67], patients receiving nivolumab and having HCC tumor tissues with inflammatory signature GEP consisting of CD274 (PD-L1), CD8α, LAG3, and STAT1, had improved objective response rate (ORR) and OS, suggesting the possibility of a relationship between underlying inflammation within the tumor environments and improved clinical outcomes. Exploratory analysis of the GO30140 study demonstrated that T-effector gene (GZM, PRF1, and CXCL9) signatures were associated with improved responses and longer PFS in patients treated with atezolizumab and bevacizumab [59].

### 3.3. Tumor Mutational Burden and High Microsatellite Instability

Tumor mutational burden (TMB) and microsatellite instability (MSI) are indirect indices of tumor antigenicity resulting from somatic tumor mutations, and these were most extensively studied for their role as predictive biomarkers in anti-PD-1 therapy. Based on KEYNOTE-158, the US FDA granted accelerated approval to pembrolizumab for the treatment of unresectable or metastatic tumor mutational burden-high (TMB-H) (≥10 mutations/megabase (mut/Mb)) solid tumors in adult and pediatric patients [71]. However, patients with HCC were not included in this study, and TMB was not high in HCC compared to that in melanoma or lung cancer [72]; moreover, TMB was not proven to be very predictive of ICI response in HCC [73]. Exploratory analysis of the GO30140 study demonstrated that TMB is unable to predict the response or PFS in patients with HCC treated with atezolizumab and bevacizumab [59]. Moreover, the phenotype MSI-high or d-MMR is very rare in HCC, with an incidence of approximately 1% [73,74]. In addition, studies have shown that it is mainly found in the early stage rather than the late stage. Therefore, as of now, routine MSI test is not considered informative in HCC.

### 3.4. WNT/β-Catenin

Mutations in *CTNNB1*, the gene responsible for encoding beta-catenin, and other alterations that affect the Wnt/beta-catenin signaling pathway are commonly found in HCC [75,76,77,78]; they are detected in approximately one-third of HCC tumors. Studies suggested that *CTNNB1* (β-catenin) mutations and consequent activation of the Wnt/β-catenin pathway could be responsible for the scarcity of immune cells in the tumor microenvironment and hence, poor clinical response to ICI [79,80]. In a genetically engineered mouse model of melanoma with constitutively active β-catenin, the latter was shown to reduce CCL4 expression, which is important for recruiting dendritic cells and, consequently, T-cells into the tumor microenvironment (TME) [79]. The mechanism by which β-catenin reduces CCL4 expression is associated with the induction of transcriptional repressor ATF3 and its binding to the CCL4 promoter [79,81,82]. The immune evasion mechanism was reproduced in an engineered HCC mouse model in which β-catenin was constantly activated; aberrant β-catenin activation resulted in increased resistance to anti-PD-1 therapy [83]. Harding et al. reported that alterations in WNT/β-catenin signaling are associated with lower disease control rate (DCR), shorter median progression-free survival (PFS), and shorter median OS in patients with advanced HCC treated with ICI [84]. Hong et al. also showed that only non-responders to pembrolizumab exhibited somatic mutations in *CTNNB1* [85]. Haber et al., on the other hand, reported that there was no association between the overall immune infiltrate or *CTNNB1* mutations and response [86]. According to the immune-specific class of HCC defined by Montironi et al., one-third was classified as inflamed tumor with Wnt/β-catenin pathway activation, and the remaining were classified as non-inflamed tumors [63]. The discordant results of Wnt/β-catenin pathway activation on its predictive potential in HCC suggest the need for further analysis.

### 3.5. Other Gene Signatures Associated with Adverse Clinical Outcomes

The biomarker study [87] with tislelizumab, an anti-PD-1 monoclonal antibody, was given to patients with advanced HCC previously treated with sorafenib (NCT02407990 and NCT04068519), and it was demonstrated that non-responders had elevated expression of genes related to angiogenesis (TEK, KDR, HGF, and EGR1), immune exhaustion (CD274, CTLA-4, TIGIT, and CD96), and cell cycle (E2F7, FOXA1, and FANCD2), compared to responders. Exploratory analysis of the GO30140 study demonstrated that gene expression related to Notch pathway activation (i.e., high expression of HES1) was associated with a lack of response and shorter PFS in patients treated with atezolizumab and bevacizumab [59].

### 3.6. Circulating Biomarkers

Unlike that in lung cancer or melanoma, studies on circulating biomarkers for immunotherapy in HCC are limited. Feun et al. reported that, among the 11 cytokines and chemokines that were tested in 24 patients with unresectable HCC and receiving pembrolizumab, only baseline TGF-β cytokine level in peripheral blood was significantly higher in non-responders than in responders [88]. Winogrand et al. reported the relevance between the presence of PD-L1^+^ circulating tumor cells (CTCs) and favorable immunotherapy outcome (*n* = 10); however, it was also a negative prognostic biomarker and an overall survival predictor (*n* = 87) [89]. Additional verification would still be required to support the small-scale studies before their incorporation as biomarkers in immunotherapy.

### 3.7. Anti-Drug Antibody against Atezolizumab

Humanized antibodies could be immunogenic and induce undesirable anti-drug antibody (ADA) responses upon administration [18,90,91]. ADAs are known to interfere with the action of a therapeutic antibody by affecting drug clearance and serum concentration [91,92] or by neutralization. ICIs were also shown to generate ADA responses in patients with cancer [91,93,94,95]. Among the various ICI antibodies, atezolizumab has the highest incidence rate of ADA (29.8%) compared to others (around 5% to 10%) [18,90,91,96]. The results of the IMbrave 150 study showed that the incidence of atezolizumab ADA reached 29.6% in patients with HCC at one or more timepoints following atezolizumab–bevacizumab treatment [97]. Although ADA-negative patients had improved OS, ADA-positive ones showed a similar OS with atezolizumab plus bevacizumab vs. sorafenib treatment (HR of ADA-positive patients vs. those of sorafenib was 0.96 (95% CI, 0.621–1.4184)). To date, however, there is no available method to predict which drug may induce ADAs, and there is no FDA-approved commercial test yet to identify the patients who may develop ADA after atezolizumab treatment [98]. Data to guide treatment decisions in patients who develop ADAs are still unavailable. Therefore, a study that can evaluate the overall effect of ADA would be appropriate in the future.

**Table 2 cancers-14-03213-t002:** Translational biomarkers.

Marker	Assay	Treatment	N	Findings Associated with Clinical Response	Reference
**TIL Based Biomarkers**					
Baseline CD3^+^ or CD8^+^ TILs	IHC	Nivolumab	189 (CD3) 192 (CD8)	CD3^+^ or CD8^+^ TILs exhibited a trend towards improved OS	Sangro et al. J. Hep. 2020 [65]
CD3^+^ or CD8^+^ TILs after Treatment	IHC	Tremelimumab with RFA or TACE	9	Responder had higher CD3^+^ or CD8^+^ TILs than non-responder	Duffy et al. J. Hep. 2017 [61]
**Sequencing based biomarkers**					
T-effector signature (*GZM, PRF1, CXCL9*)	RNA seq	Atezolizumab–Bevacizumab	90	Associated with response and longer PFS	Zhu et al. Cancer Res. 2020 [69]
Baseline inflammation signature of tumor	RNA seq	Nivolumab	37	Inflammatory signature consisting of CD274 (PD-L1), CD8A, LAG3, and STAT1 was associated with both improved objective response rate and OS.	Sangro et al. J. Hep. 2020 [65]
WNT/β-catenin	NGS	Immune checkpoint inhibitors	31	Activating alteration of WNT/β-catenin signaling was associated with lower DCR, shorter median PFS, and shorter median OS	Harding et al. Clin. Cancer Res. 2019 [82]
WNT/β-catenin	NGS	Pembrolizumab	60	Somatic mutations in CTNNB1 were found only in non-responders	Hong et al. Genome Med. 2022 [83]
Angiogenesis, Immune exhaustion, cell-cycle gene signatures	NGS	Tislelizumab	41	Non-responders had elevated angiogenesis, immune exhaustion, and cell-cycle gene signature than responders	Hou et al. J. ImmunoTher. Cancer. 2020 [85]
TCR signaling	RNA seq	Pembrolizumab	60	Responders demonstrated T cell receptor (TCR) signaling activation with expressions of MHC genes	Hong et al. Genome Med. 2022 [83]
Notch pathway activation genes	RNA seq	Atezolizumab–Bevacizumab	90	Associated with lack of response and shorter PFS	Zhu et al. Cancer Res. 2020 [69]
TMB	WES	Atezolizumab–Bevacizumab	73	Not associated with response or PFS	Zhu et al. Cancer Res. 2020 [69]
**Circulating biomarkers**					
plasma TGF-β levels	ELISA	Pembrolizumab	24	High baseline plasma TGF-β levels (≥200 pg/mL) significantly associated with unfavorable outcomes	Feun et al. Cancer 2019 [86]
Anti-drug antibody (ADA)	ELISA	Atezolizumab–Bevacizumab	336	While patients with ADA− had an improved OS, those with ADA+ had a similar OS with Ate/Bev vs. sorafenib	Galle et al. Cancer Res. 2021 [95]
PD-L1^+^CTCs	Immunocytochemistry	PD-1 blockade	10	PD-L1^+^CTCs were associated with favorable immunotherapy outcome	Winogrand et al. Hepatol. Commun. 2020 [87]

## 4. Conclusions

We reviewed the data on ICI biomarkers obtained from recent pivotal studies on HCC (Figure 1). Although several potential candidates were evaluated for predicting response to ICI treatment, there is currently no standard biomarker for ICI-treated patients with HCC. Since tissue biopsy is not mandatory for the diagnosis of HCC, the discovery of predictive biomarkers by tumor tissue analyses is limited compared to that in other solid cancers. Although PD-L1 expression in tumor tissues is known to be a predictive marker for multiple cancer types, its clinical use is less clear in HCC due to the less clear-cut association between PD-L1 expression and responders to anti-PD-1/PD-L1. In HCC, the overall picture is more complex than in other solid tumors due to the unique environment involving hepatitis and/or cirrhosis, which constantly interacts with the host’s immune system. Considering the complexity of predicting ICI treatment response in HCC, an integrative multi-parameter approach combining histopathology, imaging, and immune features would need to be applied as a novel strategy. Since immunotherapy has become the new standard of care in HCC, and various biomarker studies are being conducted in parallel, personalized therapy through a biomarker-based approach is expected to improve patient survival outcomes in the future.

## Figures and Tables

**Figure 1 cancers-14-03213-f001:**
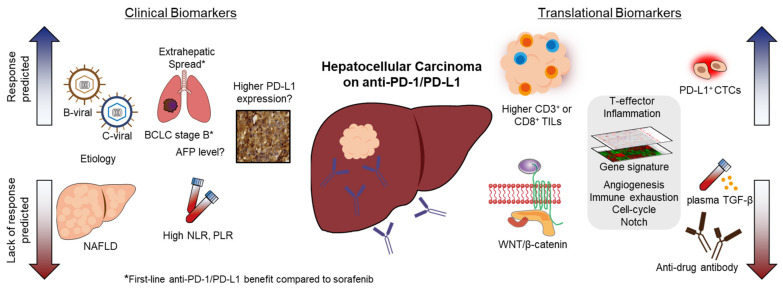
Clinical and translational biomarkers to predict the response and lack of response of immune checkpoint inhibitor treatment in hepatocellular carcinoma.

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
