# Peer review of "Could We Predict the Response of Immune Checkpoint Inhibitor Treatment in Hepatocellular Carcinoma?"

_cancers, 2022, doi:10.3390/cancers14133213_

Round 1

Reviewer 1 Report

Lee et al reported a comprehensive review of putative predictive biomarkers to ICIs in HCC. Since many advances have recently  been made in this setting, this paper could give a contribution to this field. The manuscript is well written and comprehensible.

In have only minor comments:

- I would suggest to specify that the review investigates also predictive markers to the combination of anti PD-1 and anti VEGF, other than to anti PD-1/PD-L1 inhibitors.

- In the paragraph 2.2 it is not clear the definition of "hepatic tumors".  Furthermore, it would be more accurate reporting if the correlation of metastic pattern is correlated to ICIs monotherapy or combination therapies. 

- In the paragraph 3.1 two sentences start with the first name of papers' first author name. Please replace them with their corresponding surnames.

- In the paragraph 3.3 it is reported that MSI-high HCC show poor response to ICIs. However, it seems arduous to draw these conclusions from the paper reported in the reference 73.

Author Response

Reviewer 1

Lee et al reported a comprehensive review of putative predictive biomarkers to ICIs in HCC. Since many advances have recently been made in this setting, this paper could give a contribution to this field. The manuscript is well written and comprehensible.

We thank the reviewer for the thoughtful comments.

I have only minor comments:

- I would suggest to specify that the review investigates also predictive markers to the combination of anti PD-1 and anti VEGF, other than to anti PD-1/PD-L1 inhibitors.

Response: Thank you for the comment. We hope the reviewer will understand that our manuscript mainly reviewed the predictive markers to the regimens containing anti-PD-1/PD-L1 inhibitors. Only one trial tested combination of anti-PD-L1 and anti-VEGF in our review (IMbrave150). There are trials upcoming that tests anti-PD-1+anti-VEGF combinations such as atezolizumab plus cabozatinib (COSMIC-312 trial) or pembrolizumab plus lenvatinib (LEAP-002) regimens, and these trials are not reviewed in our review.

- In the paragraph 2.2 it is not clear the definition of "hepatic tumors".  Furthermore, it would be more accurate reporting if the correlation of metastic pattern is correlated to ICIs monotherapy or combination therapies. 

Response: Thank you for the comment. We rephrased "hepatic tumors” into “intra-hepatic tumors”.

Furthermore, it would be more accurate reporting if the correlation of metastic pattern is correlated to ICIs monotherapy or combination therapies. 

Response: This is valid point. We have revised the sentences to clarify that the poorer response to intra-hepatic tumors are confined to immunotherapy monotherapy.

…Antitumor immune response to ICIs differs in an organ-specific manner [38], and liver metastasis is associated with poor response to immunotherapy monotherapy. Accordingly, intra-hepatic tumors of HCC have been reported to possibly be less responsive to immunotherapy monotherapy than extrahepatic lesions [39,40].

- In the paragraph 3.1 two sentences start with the first name of papers' first author name. Please replace them with their corresponding surnames.

Response: We reviewed the representative citation and formatting guidelines. Since those guidelines all recommend to use the first author's name before “et al.”, we hope the reviewer to understand that we used the first author name.

In Modern Language Association (MLA) style guide, they recommend to use only the first author's last name and put “et al.".

…For a source with three or more authors, list only the first author’s last name, and replace the additional names with et al....

In American Psychological Association (APA) citation and format style guide (7th edition), they also recommend to use only the first author's last name and put “et al.”

…A WORK BY THREE OR MORE AUTHORS
List only the first author’s name followed by “et al.” in every citation, even the first, unless doing so would create ambiguity between different sources. …

In The Chicago Manual of Style (17th edition), they also recommend to use only the first author's last name and put et al. when there are four or more authors.

- In the paragraph 3.3 it is reported that MSI-high HCC show poor response to ICIs. However, it seems arduous to draw these conclusions from the paper reported in the reference 73.

Response: Thank you for the comment. We agree and we will remove the phrase that MSI-high is a poor predictive indicator of ICI response in HCC.

…In addition, studies have shown that it is mainly found in the early stage rather than the late stage, and MSI-high is a poor predictive indicator of ICI response in HCC [73]. Therefore, routine MSI test is not considered informative in HCC.
à
…In addition, studies have shown that it is mainly found in the early stage rather than the late stage. Therefore, as of now, routine MSI test is not considered informative in HCC.

Reviewer 2 Report

The manuscript, "Could we predict the response and lack of response of immune checkpoint inhibitor treatment in hepatocellular carcinoma?" by Lee and colleagues is a review article that summarizes the various factors that affect clinical responses to immune checkpoint inhibitor treatment for HCC. In general this is a well done review article, with good organization, comprehensive literature review that is described with the appropriate amount of detail and conclusions that are well reasoned out. This is a good addition to the field and is expected to appeal to a broad readership comprising clinical and non-clinical investigators. It is also a good fit for the scope of Cancers. I don't have any concerns with the study and am happy to recommend it for publication.

Author Response

We thank the reviewer for the encouraging comment.

Reviewer 3 Report

The manuscript "Could we predict the response and lack of response of immune checkpoint inhibitor treatment in hepatocellular carcinoma?" adds knowledge to the field and presents potentially interesting findings.

Nevertheless, some questions should be addressed in order to improve its scientific quality:

 It is not necessary that the title includes "lack of response".

- The Introduction and the Discussion provide sufficient information to understand the state-of-the-art and citations are appropiate

- The text is not well organized and English language should be improved.

- Tables are useful.   - Links provided in the Tables are relevant.   - I would suggest to add a summary Figure.   

Author Response

Reviewer 3

The manuscript "Could we predict the response and lack of response of immune checkpoint inhibitor treatment in hepatocellular carcinoma?" adds knowledge to the field and presents potentially interesting findings.

Nevertheless, some questions should be addressed in order to improve its scientific quality:

-  It is not necessary that the title includes "lack of response".

Response: Thank you for the comment. We agree and we removed the phrase "lack of response”.

- The Introduction and the Discussion provide sufficient information to understand the state-of-the-art and citations are appropriate

- The text is not well organized and English language should be improved.

Response: We have We will revisit our English with editorial team.

- Tables are useful.  

- Links provided in the Tables are relevant.  

- I would suggest to add a summary Figure.   

Response: We hope the reviewer to find summary figure is in the Figure 1.
